# Relationship between Vitamin Intake and Resilience Based on Sex in Middle-Aged and Older Japanese Adults: Results of the Shika Study

**DOI:** 10.3390/nu14235042

**Published:** 2022-11-27

**Authors:** Kuniko Sato, Fumihiko Suzuki, Hiromasa Tsujiguchi, Akinori Hara, Takayuki Kannon, Sakae Miyagi, Keita Suzuki, Masaharu Nakamura, Chie Takazawa, Aki Shibata, Hirohito Tsuboi, Yukari Shimizu, Thao Thi Thu Nguyen, Tadashi Konoshita, Yasuki Ono, Koichi Hayashi, Atsushi Tajima, Hiroyuki Nakamura

**Affiliations:** 1Department of Clinical Cognitive Neuroscience, Graduate School of Medical Science, Kanazawa University, Kakuma-machi, Kanazawa 920-1192, Japan; 2Department of Hygiene and Public Health, Faculty of Medicine, Institute of Medical, Pharmaceutical and Health Sciences, Kanazawa University, Kanazawa 920-8640, Japan; 3Community Medicine Support Dentistry, Ohu University Hospital, Koriyama 963-8611, Japan; 4Department of Hygiene and Public Health, Graduate School of Medical Science, Kanazawa University, 13-1 Takaramachi, Kanazawa 920-8640, Japan; 5Department of Public Health, Graduate School of Advanced Preventive Medical Sciences, Kanazawa University, 13-1 Takaramachi, Kanazawa 920-8640, Japan; 6Department of Biomedical Data Science, School of Medicine, Fujita Health University, 1-98 Dengakugakubo, Kutsukake-cho, Toyoake 470-1192, Japan; 7Innovative Clinical Research Center, Kanazawa University, 13-1 Takaramachi, Kanazawa 920-8641, Japan; 8Graduate School of Human Nursing, The University of Shiga Prefecture, 2500 Hassaka-cho, Hikone 522-8533, Japan; 9Department of Nursing, Faculty of Health Sciences, Komatsu University, 14-1 Mukaimotorimachi, Komatsu 923-0961, Japan; 10Faculty of Public Health, Haiphong University of Medicine and Pharmacy, Ngo Quyen, Hai Phong 180000, Vietnam; 11Third Department of Internal Medicine, University of Fukui Faculty of Medical Sciences, 23-3 Matsuoka Shimoaizuki, Eiheiji-cho, Yoshida-gun, Fukui 910-1193, Japan; 12Department of Neuropsychiatry, Graduate School of Medicine, Hirosaki University, 1 Bunkyocyo, Hirosaki 036-8224, Japan; 13Department of Food Sciences and Nutrition, School of Human Environmental Sciences, Mukogawa Women’s University, 6-46, Ikebiraki-cho, Nishinomiya 663-8558, Japan; 14Advanced Preventive Medical Sciences Research Center, Kanazawa University, 1-13 Takaramachi, Kanazawa 920-8640, Japan; 15Department of Bioinformatics and Genomics, Graduate School of Advanced Preventive Medical Sciences, Kanazawa University, 13-1 Takaramachi, Kanazawa 920-8640, Japan

**Keywords:** vitamin intake, resilience, middle-aged and older Japanese adults, cross-sectional study, sex

## Abstract

Epidemiological studies reported that resilience, generally regarded as the ability to manage stress in the face of adversity, correlates with mental health in middle-aged and older adults. Currently, there is limited information on eating habits that affect resilience. Therefore, this cross-sectional study investigated the relationship between vitamin intake and resilience based on sex in community-dwelling middle-aged and older individuals in Shika town, Ishikawa Prefecture, Japan. A total of 221 participants (106 men and 115 women) aged 40 years or older were included in the analysis. We assessed vitamin intake and resilience using a brief-type self-administered diet history questionnaire (BDHQ) and the resilience scale (RS), respectively. A two-way analysis of covariance (ANCOVA) revealed that higher intakes of β-carotene and vitamin K were associated with higher RS in women, but not in men. Furthermore, a multiple logistic regression analysis stratified by sex showed that β-carotene and vitamin K were significant independent variables for RS only in women. The present study suggests that higher intakes of β-carotene and vitamin K were associated with higher resilience among middle-aged and older women. The results obtained demonstrate that β-carotene and vitamin K intakes may enhance resilience by strengthening stress tolerance.

## 1. Introduction

Resilience is a psychological aspect of individuals that refers to inner perseverance, equanimity, meaningfulness, self-reliance, and existential aloneness [1,2]. A positive correlation between resilience and stress perception and a negative correlation between resilience and depressive symptoms were reported [1], suggesting that resilience may be an important factor for mental health. With aging, middle-aged and older adults become more sensitive to the impact of environmental factors, are more susceptible to changes strongly associated with mental and physical medical conditions, and may face serious challenges, such as health deterioration and the loss of loved family members [3,4]. Although large epidemiological studies on elderly subjects reported a relationship between higher resilience and better health/longevity, the factors contributing to this relationship have not been elucidated in detail [4,5,6,7,8]. Since evidence to support resilience as a protective factor against mental and physical symptoms in middle-aged and older adults is limited [9,10], strategies to improve mental health in middle-aged and older adults are still being investigated [4,11,12,13].

A proper diet in middle-aged and older adults has been identified as an essential factor affecting physical health [14,15]. Epidemiological studies on the relationship between food intake and resilience showed that high resilience was associated with dietary diversity [16] and a high fish intake [17], while low resilience correlated with a low intake of fried food [14]. In women, high resilience has been associated with higher intakes of vegetables, fruits, fish, dietary fiber, and nuts [18] and with the intake of bifidobacteria [19]. Therefore, the relationship between diet and resilience is expected to have a specific direction, i.e., a Western diet with a high fat intake conflicting with high resilience [20,21]. Furthermore, there may be sex differences that have not yet been examined in detail.

Recent studies in Western countries indicated that lower intakes of vitamins, minerals, and other nutrients than the recommended amounts often led to inherently preventable health issues in the elderly [22]. Furthermore, epidemiological studies indicated a relationship between vitamin deficiencies and depressive symptoms [23] or lower QOL [24], particularly in women. Research that focuses on vitamin intake to alleviate mental instability in women is particularly significant due to concerns regarding chronic stress among women, who play diverse social roles in today’s complex society [25]. Vitamin A and β-carotene intakes may be inversely associated with depression [26] and individuals with a higher vitamin K intake may be less likely to develop depressive symptoms [27]. However, limited information is currently available on the relationship between vitamin intake and resilience. Nevertheless, a low vitamin D intake has been associated with low resilience [28] and vitamin D has potential as a marker of resilience against potentially fatal disease lethality [29].

Therefore, we herein performed a cross-sectional study to elucidate the relationship between vitamin intake and resilience based on sex in 221 middle-aged and older individuals in Shika town, Ishikawa Prefecture, Japan.

## 2. Materials and Methods

### 2.1. Study Population

We utilized the data from the Shika study. The Shika study is a continuous, community-based investigation that aim to develop advanced preventive measures for lifestyle-related diseases [23,30,31]. This survey consists of a questionnaire and, if necessary, data from health checkups. Furthermore, the Shika study examines the relationship between nutrition and depressive symptoms in the elderly [23,30,31]. Shika Town is in a rural area of Ishikawa Prefecture, Japan, with 19,206 inhabitants. The climate is subtropical, and the major industries are electronics manufacturing, retail, and health and social services. [23,30,31]. Investigation was conducted from January 2020 to April 2021 in four model districts in Shiga Town (Higashi Masuho, Horimatsu, Togi, and Tsuchida). Data were obtained from individuals 40 years and older residing in each model district.

We distributed invitation letters to 397 people, of whom 328 (82.6%) participated in the survey. Among the participants who responded, 106 men and 115 women, a total of 221 participants, excluding those taking antidepressants and with incomplete responses, were included in the analysis. All participants provided written informed consent for inclusion before they participated in the study. The present study was conducted following the Declaration of Helsinki, and the protocol was approved by the Ethics Committee of Kanazawa University (No. 1491).

### 2.2. Evaluation of Nutritional Intake

Nutrient intake was assessed using a brief, self-administered diet history questionnaire (BDHQ) [32,33]. BDHQ gathers information on average food intake over the last month to estimate the habitual intake of nutrients from ordinary foods. The questionnaire asked about the intakes of 58 different foods, alcoholic and non-alcoholic beverages, the daily intake of rice and miso soup, and recipes with regular fish meat from the typical diet. Based on BDHQ, we used data on the intakes of energy and 13 vitamins: retinol, β-carotene (“β-carotene” stands for “beta-carotene equivalent”), vitamin D, α-tocopherol, vitamin K, vitamin B_1_, vitamin B_2_, niacin, vitamin B_6_, vitamin B_12_, folic acid, pantothenic acid, and vitamin C. The validity of BDHQ in Japanese populations was previously demonstrated [32,33]. The use of dietary supplements may lead to an excessive intake of micronutrients, such as minerals and vitamins, which may affect the interpretation of the results. To avoid this bias, the intake of minerals or vitamins from supplements was not included in calculations. Participants who reported a total energy intake of less than 600 kcal/day (half the energy intake required for the low physical activity category) or more than 4000 kcal/day (1.5 times the energy intake required for the moderate physical activity category) were excluded from the analysis due to potential bias.

### 2.3. Resilience Scale

The Resilience Scale (RS), which has proven reliability and validity, was used to assess resilience [2]. The reliability and validity of the RS Japanese version have also been confirmed [1]. Resilience is a self-report measure that reflects five characteristics (purpose, perseverance, self-reliance, equanimity, and existential aloneness) from 25 items. For example, “I can get through difficult times because I’ve experienced difficulty before” and “I can usually find something to laugh about” are included and scored from 1 = strongly disagree to 7 = strongly agree, with possible scores ranging between 25 and 175, which is then evaluated as “RS.” Higher scores indicate higher resilience.

### 2.4. Other Variables

We used the medical checkup data conducted as the Shika study. Data on age, sex (0: men; 1: women), body mass index (BMI: calculated based on the standard formula kg/m^2^ using measured height and weight), current smoking and drinking status (0: no, 1: yes), duration of education (0: <12, 1: >13 years), and “with occupation” (0: no, 1: yes) were collected.

### 2.5. Statistical Analysis

Continuous variables were described as mean and standard deviation (SD), while categorical variables were expressed as numbers (N) and percentages (%). The Kolmogorov–Smirnov and Shapiro–Wilk tests were utilized to evaluate normality. The characteristics of and differences in each participant according to sex were assessed using the Student’s *t*-test (continuous variable) and χ^2^ test. RS was classified as either the low RS group (RS ≤ 117) or high RS group (RS ≥ 118) using the median value.

Vitamins based on BDHQ were used as the dependent variable for each sex, adjusted for age, BMI, the smoking status, drinking status, years of education (<12 or >13 years), and “with occupation” as covariates, and the interaction between sex and the high and low RS groups was analyzed using a two-way analysis of covariance (two-way ANCOVA). Multiple comparisons were made using the Bonferroni method for items that showed an interaction between sex and high and low RS. A multiple logistic regression analysis was performed using β-carotene or vitamin K and age, BMI, the smoking status, drinking status, years of education, and “with occu-pation” as explanatory variables, with high and low RS as objective variables, and as an explanatory variable.

A type I error of 0.05 was used for all analyses, with *p*-values between 0.05 and 0.10 being considered to indicate borderline significance. IBM SPSS Statistics version 25 for Windows (IBM, Armonk, NY, USA) was used for statistical analyses.

## 3. Results

### 3.1. Participant Characteristics

Table 1 shows the participants characteristics. Among 221 participants, the mean age of 66.6 years (SD = 10.5) in 106 men was not significantly different that of 64.6 years (SD = 9.8) in 115 women. BMI was significantly higher in men than in women (*p* < 0.001). The percentages of current smokers (*p* = 0.002) and current drinkers (*p* < 0.001) were significantly higher among men than among women. The percentage of women with >13 years of education was significantly higher than that of men (*p* < 0.020). The intakes of β-carotene (*p* < 0.001), α-tocopherol (*p* < 0.001), vitamin K (*p* = 0.002), vitamin B_1_ (*p* < 0.001), vitamin B_2_ (*p* < 0.001), niacin (*p* = 0.001), vitamin B_6_ (*p* < 0.001), folic acid (*p* = 0.007), pantothenic acid (*p* < 0.001), and vitamin C (*p* = 0.004) were significantly higher in women than in men.

### 3.2. Characteristics of RS Groups

The characteristics of participants classified into the two RS groups are shown in Table 2. The intakes of β-carotene (*p* < 0.001), α-tocopherol (*p* < 0.001), vitamin K (*p* = 0.002), vitamin B_1_ (*p* < 0.001), vitamin B_2_ (*p* < 0.001), niacin (*p* = 0.001), vitamin B_6_ (*p* < 0.001), folic acid (*p* = 0.007), pantothenic acid (*p* < 0.001), and vitamin C (*p* = 0.004) were significantly higher in the high RS group than in the low RS group.

### 3.3. Relationship between Vitamin Intake in RS Groups by Sex

Table 3 shows the results of the two-way ANCOVA adjusted for age, BMI, current smoking and drinking status, education, and occupation for the relationships between the two RS groups and the two sex groups regarding vitamin intake. The two RS groups were subclassified according to sex. Among men, 56 were in the low-RS group and 50 in the high-RS group. Among women, 59 were in the low-RS group and 56 in the high-RS group. The main effects of vitamin intake in the two sex groups, namely, β-carotene (*p* = 0.002), α-tocopherol (*p* < 0.001), vitamin K (*p* = 0.010), vitamin B_1_ (*p* < 0.001), vitamin B_2_ (*p* < 0.001), niacin (*p* = 0.001), vitamin B_6_ (*p* < 0.001), folic acid (*p* = 0.012), pantothenic acid (*p* < 0.001), and vitamin C (*p* = 0.026), were significantly higher in women than in men. Interactions between the two sex groups and two RS groups were observed for Retinol (*p* = 0.009), β-carotene (*p* = 0.045), and Vitamin K (*p* = 0.033). Multiple comparisons using the Bonferroni method showed that the intake of β-carotene was significantly higher in women than in men in the high-RS group (*p* < 0.001), whereas there was no sex difference in the low-RS group. Similarly, in the high-RS group, the intake of vitamin K was significantly higher in women than in men (*p* = 0.001), whereas there was no sex difference in the low-RS group. Therefore, the intakes of β-carotene and vitamin K were significantly higher in women in the high-RS group than in those in the low-RS group.

### 3.4. Relationship between β-Carotene and RS Stratified by Sex

Table 4 shows the results of the multiple logistic regression analysis stratified by sex, with high and low RS as objective variables and β-carotene as an explanatory variable. Among women, β-carotene (β = 0.001, *p* = 0.001) was a significant explanatory variable for high and low RS in all four models with different covariates. Among men, there was no correlation between β-carotene and RS in all models. This relationship was also confirmed in the analysis with β-Carotene and sex as interaction terms (Appendix A).

### 3.5. Relationship between Vitamin K and RS Stratified by Sex

Table 5 shows the results of the multiple logistic regression analysis with vitamin K as the explanatory variable, using the same analysis method as that in Table 4. Among women, vitamin K (β = 0.007, *p* = 0.003) was a significant explanatory variable for high and low RS in all four models with different covariates. Among men, no explanatory variable correlated with the two RS groups. This relationship was also confirmed in the analysis using vitamin K and sex as interaction terms (Appendix A).

## 4. Discussion

The results of the present study indicated that β-carotene and vitamin K intakes were associated with higher resilience in middle-aged and older women.

We found that the significant main effects of resilience on the intakes of major vitamins, such as vitamins A, B, C, D, E, and K, niacin, folic acid, and pantothenic acid, were significantly higher in the high-RS group than in the low-RS group. Previous studies reported a lower risk of depression and depressive symptoms with the consumption of antioxidants, including green tea polyphenols and isoflavonoids, in a balanced diet, such as the Mediterranean diet or that including certain foods, such as fish, fresh vegetables, and fruits, while high-fat Western diets and sugar-sweetened beverages were associated with a higher risk of depression and depressive symptoms [34,35]. Recent studies, including animal studies, showed that various phytochemicals, such as the peel of immature citrus, possess biological functions that facilitate the development of stress resilience, and the anti-inflammatory effects of phytochemicals may play a key role in protection against psychosocial stress [36]. Curcumin, a biologically active polyphenol compound found in the turmeric plant [37], and the citrus flavonoid hesperidin [38] have been reported to promote resilience to chronic stress. The present results, showing that the main vitamins in the high-RS group were significantly higher than those in the low-RS group, suggest a relationship between vitamin intake and resilience, which may also be a protective factor against physical and mental illnesses in middle-aged and older adults.

The two-way ANCOVA showed a significant interaction between sex and resilience with β-carotene and vitamin K intakes. Vitamin intake may be a marker of resilience against potentially fatal disease lethality [29] and has been implicated in stress management by women [25], depressive symptoms [23], and quality of life [24]. Carotenoids, used as vitamin A precursors, have been associated with processes that positively impact human health [39,40]. Furthermore, epidemiological studies demonstrated that α-carotenoid and β-carotenoid intakes, generally resulting from the high consumption of fruits and vegetables, may be inversely associated with depressive symptoms in late midlife women [39,40]. A cross-sectional study using data from the Study of Women’s Health Across the Nation on health among women in late middle age in the United States also suggested that α-carotene and β-carotenoid intakes were inversely associated with depressive symptoms in late midlife women; therefore, depression indicating potential pathophysiological pathways may involve inflammation and oxidative stress and may be prevented by the anti-depressant effects of carotenoids [41].

The two-way ANCOVA adjusted for age, BMI, current smoking and drinking status, education, and occupation revealed that the significant main effects of sex on the major vitamins, including vitamins A, B, C, E, and K, niacin, folic acid, and pantothenic acid, were significantly higher in women than in men. Regarding the relationship between vitamin intake and sex, a lower vitamin intake and increased risk of folate deficiency were reported for adults and older European women [42]. In a cohort study, Marques-Vidal et al. (2015) [43] examined dietary intake by sex and educational attainment among Swiss adults and showed that carotene and vitamin D intakes in both men and women were higher between 1993 and 1999 for those with a higher education than for those with a lower education, and this difference decreased between 2006 and 2012 [43]. Based on these findings [43] and the present results, the significantly higher average intakes of vitamin A and other vitamins in women than in men may be related to the percentage of highly educated individuals being significantly higher in women than in men. Taken together with the findings reported by Marques-Vidal et al. (2015) [43], one reason for the significant main effect of sex on the intake of major vitamins may be a sex difference in attitudes towards health care.

In the brain, vitamin K has been suggested to influence psychomotor behavior and cognition by preventing oxidative stress and inflammation from affecting important cellular events, such as proliferation, differentiation, senescence, and cell–cell interactions [44]. Epidemiological studies on middle-aged North Americans [27] and elderly Japanese [23] reported a negative correlation between depression and vitamin K intake. Based on these findings, the present results suggest that higher β-carotene and vitamin K intakes are associated with higher resilience in middle-aged and older women. Therefore, β-carotene and vitamin K intakes may enhance psychological stress coping and resilience. In addition to interventions such as cognitive-behavioral therapy and mindfulness [45] to enhance resilience with strong psychological and cognitive aspects, from a public health perspective, further clarification of the relationship between the daily diet and resilience, including β-carotene and vitamin K intakes, will lead to the development of new measures that will improve the mental health of middle-aged and older adults.

There are a number of limitations that need to be addressed. Since this was a cross-sectional study, we were unable to elucidate the mechanisms contributing to the relationship between vitamin intake and resilience. Due to the small sample size in this study, the relationship between RS and β-carotene or vitamin K ought to confirm in a larger sample size. Further longitudinal studies are needed to investigate causal relationships. Moreover, the present results on vitamin intake and resilience were limited by the use of a self-report questionnaire. Nevertheless, the two measures that were employed have been validated and widely used in studies on middle-aged and older adults [1,46]. In addition, although we adjusted for a number of potential confounding variables, we were unable to eliminate residual confounding variables, such as physical activity, economic income, a history of medical conditions, such as chronic disease or dementia and resulting medication use, and a history of vitamin supplement use.

## 5. Conclusions

The present results indicate that higher β-carotene and vitamin K intakes were associated with higher resilience among middle-aged and older women, but not men. The difference observed between women and men seems to be attributed to the higher education in women than in men in the present study. Further longitudinal studies with larger sample sizes are needed to investigate this causal relationship.

## Figures and Tables

**Table 1 nutrients-14-05042-t001:** Participant characteristics according to sex.

	Men (*n* = 106)		Women (*n* = 115)		*p*-Value *
	Mean (*n*)	SD (*%*)	Mean (*n*)	SD (*%*)	
Age, years	66.6	10.5	64.6	9.8	0.141
BMI, kg/m^2^	24.2	3.3	22.5	2.8	**<0.001**
Current smoker, *n*	23	21.7	8	7.0	**0.002**
Current drinker, *n*	73	68.9	46	40.0	**<0.001**
Education of more than 12 years, *n*	37	34.9	58	50.4	**0.020**
with occupation, *n*	65	61.3	65	56.5	0.469
RS (25–175)	118.5	20.5	119.2	20.7	0.821
Energy (kcal/day)	2188.72	725.55	1653.82	401.48	**<0.001**
Retinol (µg/1000 kcal)	193.88	129.27	170.78	98.84	0.135
β-Carotene (µg/1000 kcal)	1605.62	1066.15	2267.12	1530.31	**<0.001**
Vitamin D (µg/1000 kcal)	7.89	5.37	8.36	4.75	0.486
α-Tocopherol (mg/1000 kcal)	3.56	0.99	4.17	0.99	**<0.001**
Vitamin K (µg/1000 kcal)	141.93	74.29	178.34	98.23	**0.002**
Vitamin B_1_ (mg/1000 kcal)	0.37	0.09	0.45	0.11	**<0.001**
Vitamin B_2_ (mg/1000 kcal)	0.62	0.17	0.72	0.18	**<0.001**
Niacin (mg/1000 kcal)	8.90	2.49	10.03	2.71	**0.001**
Vitamin B_6_ (mg/1000 kcal)	0.64	0.16	0.74	0.200	**<0.001**
Vitamin B_12_ (µg/1000 kcal)	5.46	2.76	5.67	2.68	0.562
Folic acid (µg/1000 kcal)	160.27	59.81	184.24	70.34	**0.007**
Pantothenic acid (mg/1000 kcal)	3.15	0.66	3.62	0.73	**<0.001**
Vitamin C (mg/1000 kcal)	51.72	27.95	62.87	29.67	**0.004**

* Student’s *t*-test for continuous variables and chi-square test for categorical variables was applied, respectively. (*p*-values less than 0.05 are highlighted in bold). Abbreviations: SD, standard deviation; BMI, body mass index; RS, resilience scale.

**Table 2 nutrients-14-05042-t002:** Differences between RS ≤117 (*n* = 115) and ≥118 (*n* = 106) groups.

	RS ≤ 117 (*n* = 115)		RS ≥ 118 (*n* = 106)		*p*-Value ***
	Mean (*n*)	SD (*%*)	Mean (*n*)	SD (*%*)	
Age, years	64.5	9.9	66.6	10.4	0.130
Sex (men) (*n*, %)	56	49.1	50	47.2	0.821
BMI, kg/m^2^	23.2	3.3	23.5	3.1	0.473
Current smoker, *n*	15	13.0	16	15.1	0.661
Current drinker, *n*	64	55.7	55	51.9	0.575
Education of more than 12 years, *n*	50	43.5	45	42.5	0.878
with occupation, *n*	67	58.3	63	59.4	0.859
Energy (kcal/day)	1904.81	674.66	1916.41	597.89	0.89
Retinol (µg/1000 kcal)	169.43	104.05	195.34	124.46	0.100
β-Carotene (µg/1000 kcal)	1577.14	966.51	2354.18	1605.52	**<0.001**
Vitamin D (µg/1000 kcal)	7.33	4.35	9.00	5.600	**0.015**
α-Tocopherol (mg/1000 kcal)	3.63	0.96	4.15	1.04	**<0.001**
Vitamin K (µg/1000 kcal)	141.66	74.71	181.72	98.93	**0.001**
Vitamin B_1_ (mg/1000 kcal)	0.39	0.10	0.44	0.10	**<0.001**
Vitamin B_2_ (mg/1000 kcal)	0.64	0.16	0.71	0.19	**0.003**
Niacin (mg/1000 kcal)	8.83	2.39	10.19	2.77	**<0.001**
Vitamin B_6_ (mg/1000 kcal)	0.64	0.18	0.74	0.19	**<0.001**
Vitamin B_12_ (µg/1000 kcal)	5.08	2.53	6.11	2.82	**0.005**
Folic acid (µg/1000 kcal)	157.21	55.08	189.60	73.52	**<0.001**
Pantothenic acid (mg/1000 kcal)	3.25	0.67	3.55	0.77	**0.002**
Vitamin C (mg/1000 kcal)	52.91	29.51	62.54	28.43	**0.014**

* Student’s *t*-test for continuous variables and chi-square test for categorical variables was applied, respectively (*p*-values less than 0.05 are highlighted in bold). Abbreviations: SD, standard deviation; BMI, body mass index; RS, resilience scale.

**Table 3 nutrients-14-05042-t003:** Interactions between sex and the resilience scale (RS) according to vitamin intake.

Vitamin	Sex	Total (*n* = 221)						*p*-Value *		
		RS ≤ 117 (*n* = 56)			RS ≥ 118 (*n* = 50)			Sex	RS	Sex × RS
		Ave	95% CI		Ave	95% CI				
			Lower	Upper		Lower	Upper			
Retinol (µg/1000 kcal)	men	154.34	123.76	184.93	217.75	185.22	250.28	0.698	0.130	**0.009**
	women	187.92	157.80	218.03	170.95	140.83	201.06			
β-Carotene (µg/1000 kcal)	men	1484.00	1147.34	1820.66	1837.98	1479.96	2196.00	**0.002**	**<0.001**	**0.045**
	women	1724.49	1393.00	2055.99	2752.97	2421.48	3084.46			
Vitamin D (µg/1000 kcal)	men	6.79	5.46	8.12	8.88	7.46	10.29	0.384	**0.021**	0.415
	women	7.97	6.66	9.29	8.98	7.67	10.29			
α-Tocopherol (mg/1000 kcal)	men	3.36	3.10	3.63	3.85	3.57	4.13	**<0.001**	**<0.001**	0.951
	women	3.92	3.66	4.18	4.39	4.13	4.64			
Vitamin K (µg/1000 kcal)	men	139.63	117.05	162.22	149.28	125.26	173.30	**0.010**	**0.003**	**0.033**
	women	148.16	125.92	170.40	205.87	183.63	228.11			
Vitamin B_1_ (mg/1000 kcal)	men	0.36	0.33	0.38	0.40	0.37	0.42	**<0.001**	**<0.001**	0.720
	women	0.42	0.39	0.44	0.46	0.44	0.49			
Vitamin B_2_ (mg/1000 kcal)	men	0.59	0.55	0.64	0.66	0.61	0.70	**<0.001**	**0.007**	0.942
	women	0.69	0.64	0.73	0.75	0.70	0.79			
Niacin (mg/1000 kcal)	men	8.21	7.52	8.91	9.47	8.73	10.21	**0.001**	**<0.001**	0.781
	women	9.41	8.73	10.10	10.86	10.17	11.54			
Vitamin B_6_ (mg/1000 kcal)	men	0.61	0.56	0.66	0.67	0.62	0.72	**<0.001**	**<0.001**	0.325
	women	0.69	0.64	0.73	0.80	0.75	0.84			
Vitamin B_12_ (µg/1000 kcal)	men	4.80	4.08	5.52	6.15	5.39	6.92	0.564	**0.004**	0.379
	women	5.34	4.64	6.05	6.07	5.36	6.78			
Folic acid (µg/1000 kcal)	men	152.98	136.41	169.56	169.03	151.41	186.66	**0.012**	**0.001**	0.184
	women	165.49	149.17	181.81	203.47	187.15	219.79			
Pantothenic acid (mg/1000 kcal)	men	3.06	2.88	3.24	3.29	3.09	3.48	**<0.001**	**0.005**	0.740
	women	3.47	3.29	3.64	3.75	3.58	3.93			
Vitamin C (mg/1000 kcal)	men	50.12	42.66	57.58	55.48	47.55	63.42	**0.026**	**0.049**	0.589
	women	57.46	50.11	64.80	66.83	59.48	74.18			

* *p*-values were calculated from a two-way analysis of covariance (*p*-values less than 0.05 are highlighted in bold). Adjusted for age, BMI, current smoker, current drinker, education, and occupation. Abbreviations: CI, confidence interval; RS, resilience scale.

**Table 4 nutrients-14-05042-t004:** Relationship between β-carotene and the RS stratified by sex.

Model	Factors	Men					Women				
		β	*p*-Value	OR	95% CL		β	*p*-Value	OR	95% CL	
					Lower	Upper				Lower	Upper
1	β-Carotene	0.000	0.052	1.000	1.000	1.001	0.001	**0.001**	1.001	1.000	1.001
2	β-Carotene	0.000	0.064	1.000	1.000	1.001	0.001	**0.001**	1.001	1.000	1.001
3	β-Carotene	0.000	0.070	1.000	1.000	1.001	0.001	**0.001**	1.001	1.000	1.001
4	β-Carotene	0.000	0.078	1.000	1.000	1.001	0.001	**0.001**	1.001	1.000	1.001

Significant estimates are in bold. Abbreviations: BMI, body mass index; RS, resilience status; OR, odds ratio, CI, confidence interval. Model 1; adjusted for age and BMI, Model 2; adjusted for age, BMI, current smoker, and current drinker. Model 3; adjusted for age, BMI, education, and occupation, Model 4; adjusted for age, BMI, current smoker, current drinker, education, and occupation.

**Table 5 nutrients-14-05042-t005:** Relationship between vitamin K and the RS stratified by sex.

Model	Factors	Men					Women				
		β	*p*-Value	OR	95% CL		β	*p*-Value	OR	95% CL	
					Lower	Upper				Lower	Upper
1	Vitamin K	0.003	0.346	1.003	0.997	1.008	0.007	**0.003**	1.007	1.002	1.011
2	Vitamin K	0.002	0.388	1.002	0.997	1.008	0.007	**0.003**	1.007	1.002	1.012
3	Vitamin K	0.002	0.453	1.002	0.997	1.008	0.007	**0.003**	1.007	1.002	1.011
4	Vitamin K	0.002	0.466	1.002	0.996	1.008	0.007	**0.003**	1.007	1.002	1.012

Significant estimates are in bold. Abbreviations: BMI, body mass index; RS, resilience status; OR, odds ratio, CI, confidence interval. Model 1; adjusted for age and BMI, Model 2; adjusted for age, BMI, current smoker, and current drinker. Model 3; adjusted for age, BMI, education, and occupation, Model 4; adjusted for age, BMI, current smoker, current drinker, education, and occupation.

## Data Availability

Data in the present study are available upon request from the corresponding author. Data are not publicly available due to privacy and ethical policies.

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
