# Peer review of "Relationship between Vitamin Intake and Resilience Based on Sex in Middle-Aged and Older Japanese Adults: Results of the Shika Study"

_nutrients, 2022, doi:10.3390/nu14235042_

Round 1

Reviewer 1 Report

The authors presented the relationship between vitamin intake and resilience in middle-aged and older Japanese adults. The topic is interesting, but the reviewer has big concerns about the results and discussion. That is, the statistics values in Table 4 are likely to be too small to show significance in the relationship between beta carotene and the RS and the discussion points seem not clear. Further comments are below.

Introduction,

The Introduction is well-written, but if the authors are going to mention the association between nutrients and depression in lines 87-88, the reviewer suggests that the authors would introduce in the first paragraph the known association between resilience and mental health including depression.

Material Methods,

As for the sentence from line 155, page 4, ‘BDHQ was used as the dependent variable for …, and the interaction between sex and the high and low RS groups was analyzed using a two-way analysis of covariance (two-way ANCOVA),’ did the authors use not BDHQ but vitamins based on BDHQ as the dependent variables?

β-carotene and vitamin K are listed as objective variables in '2.5. Statistical Analysis' (line 162, page 4), but RS is stated as the objective variable in the results section (line 222, page 7). Which is correct? And if RS is the objective variable in the logistic regression, is high RS an outcome? As for the explanatory variables, which one is 1 for each categorical variable (e.g., for gender, 1 for females, 0 for males, etc.)?

To test the sex interaction on the relationship between vitamin intake and resilience, I recommend including the interaction term between sex and vitamin (beta carotene or vitamin K) as well as vitamin and covariates in the multiple logistic regression analysis for resilience.

Is there any overlap in ‘A type I error of 0.05 was used for…’ (line 164, page 4) with the previous sentence?

Results,

I cannot well understand the sentence ‘Among men, no explanatory variable correlated with the two RS groups.’ (lines 224-225, page 7). Is it that there was no association between beta-carotene and RS in all models in men?

Discussion,

Based on the Introduction, I believe the primary aim of the study is to examine the relationship between vitamin intake and resilience and its sex interaction. Currently, the second paragraph discusses the differences between vitamin intake and sex, but I recommend discussing the main results of the study, the relationship between vitamin intake and resilience. Then, it would be better to consider how sex effect-modified its relationship and discuss its mechanism.

In the current third and fourth paragraphs of the Discussion, there seems to be mixed discussion on the relationship between nutrients and resilience and depression.

Conclusion,

I’m not sure why the authors could state higher education is attributed to the observed sex difference in the relationship between vitamin intake and resilience as a conclusion. It is recommended to state the conclusion on the primary objective, the overall study, and scientific implications.

Author Response

Reviewer 1

Comments and Suggestions for Authors

The authors presented the relationship between vitamin intake and resilience in middle-aged and older Japanese adults. The topic is interesting, but the reviewer has big concerns about the results and discussion. That is, the statistics values in Table 4 are likely to be too small to show significance in the relationship between beta carotene and the RS and the discussion points seem not clear. Further comments are below.

Comment 1

Introduction,

The Introduction is well-written, but if the authors are going to mention the association between nutrients and depression in lines 87-88, the reviewer suggests that the authors would introduce in the first paragraph the known association between resilience and mental health including depression.

Response 1

We have added the following sentence to the abstract section:

“A positive correlation between resilience and stress perception and a negative correlation between resilience and depressive symptoms were reported [1], suggesting that resilience may be an important factor for mental health.” (L61-64)

Comment 2

Material Methods,

As for the sentence from line 155, page 4, ‘BDHQ was used as the dependent variable for …, and the interaction between sex and the high and low RS groups was analyzed using a two-way analysis of covariance (two-way ANCOVA),’ did the authors use not BDHQ but vitamins based on BDHQ as the dependent variables?

Response 2

We have amended the following sentence to the Material Methods section:

“Vitamins based on BDHQ were used as the dependent variable” (L162)

Comment 3

β-carotene and vitamin K are listed as objective variables in '2.5. Statistical Analysis' (line 162, page 4), but RS is stated as the objective variable in the results section (line 222, page 7). Which is correct? And if RS is the objective variable in the logistic regression, is high RS an outcome? As for the explanatory variables, which one is 1 for each categorical variable (e.g., for gender, 1 for females, 0 for males, etc.)?

Response 3

We have amended the Statistical Analysis section as follows:

“A multiple logistic regression analysis was performed using β-carotene or vitamin K and age, BMI, the smoking status, drinking status, years of education, and “with occupation” as explanatory variables, with high and low RS as objective variables, and as an explanatory variable.” (L167-170)

Since high RS is an outcome, we have amended the following sentence to the Resilience Scale section:

“Higher scores indicate higher resilience” (L145)

To clarify the categorization, we have amended the other variables sections as follows:

“We used the medical checkups data conducted as the Shika study. Data on age, sex (0: men; 1: women), body mass index (BMI: calculated based on the standard formula kg/m2 using measured height and weight), current smoking and drinking status (0: no, 1: yes), duration of education (0: < 12, 1: >13 years), and “with occupation” (0: no, 1: yes) were collected.” (L147-151)

Comment 4

To test the sex interaction on the relationship between vitamin intake and resilience, I recommend including the interaction term between sex and vitamin (beta carotene or vitamin K) as well as vitamin and covariates in the multiple logistic regression analysis for resilience.

Response 4

We have adopted the present table form to emphasize that the relationship between vitamin intake and resilience differs by gender.

On the other hand, since your point is also very appropriate, we decided to present the tables with the interaction term as a variable in Supplementary Tables 1 and 2.

Additionally, we have added following sentence to the Results section:

“This relationship was also confirmed in the analysis with β-Carotene and sex as interaction terms (Supplementary Table 1).” (L 240-242)

“This relationship was also confirmed in the analysis with vitamin K and sex as interaction terms (Supplementary Table 2).” (L255-256)

Comment 5

Is there any overlap in ‘A type I error of 0.05 was used for…’ (line 164, page 4) with the previous sentence?

Response 5

We have removed this sentence from the Statistical Analysis section. (L171)

Comment 6

Results,

I cannot well understand the sentence ‘Among men, no explanatory variable correlated with the two RS groups.’ (lines 224-225, page 7). Is it that there was no association between beta-carotene and RS in all models in men?

Response 6

We have modified this text as follows:

“Among men, there was no correlation between β-carotene and RS in all models.” (L233-234)

Comment 7

Discussion,

Based on the Introduction, I believe the primary aim of the study is to examine the relationship between vitamin intake and resilience and its sex interaction. Currently, the second paragraph discusses the differences between vitamin intake and sex, but I recommend discussing the main results of the study, the relationship between vitamin intake and resilience. Then, it would be better to consider how sex effect-modified its relationship and discuss its mechanism.

In the current third and fourth paragraphs of the Discussion, there seems to be mixed discussion on the relationship between nutrients and resilience and depression.

Response 7

We have revised the discussion paragraphs to reorganize them and clarify the issues.

Comment 8

Conclusion,

I’m not sure why the authors could state higher education is attributed to the observed sex difference in the relationship between vitamin intake and resilience as a conclusion. It is recommended to state the conclusion on the primary objective, the overall study, and scientific implications.

Author's Reply to the Review Report (Reviewer 2)

Comments and Suggestions for Authors

This is an interesting study, given the limited number of studies concerning the relationship between vitamin intake and resilience, understood as the psychological response to environmental stress.

Response 8

We have amended the Conclusions section as follows:

“The differences observed between women and men seems to be due to the fact that women are more educated than men in this study.” (L344-346)

Supplementary Table 1. Relationship between β-carotene and sex as interaction term on RS

Model

Factors

β

p - value

OR

95%CL

Lower

Upper

1

β-carotene*sex

0.000

< 0.001

1.000

1.000

1.000

2

β-carotene*sex

0.000

< 0.001

1.000

1.000

1.000

3

β-carotene*sex

0.000

< 0.001

1.000

1.000

1.000

4

β-carotene*sex

0.000

< 0.001

1.000

1.000

1.000

Significant estimates are in bold. Abbreviations: BMI, body mass index; RS, resilience status; OR, odds ratio, CI, confidence interval. Model 1; adjusted for age and BMI, Model 2; adjusted for age, BMI, current smoker, and current drinker. Model 3; adjusted for age, BMI, education, and occupation, Model 4; adjusted for age, BMI, current smoker, current drinker, education, and occupation.

Supplementary Table 2. Relationship between vitamin K and sex as interaction term on RS

Model

Factors

β

p - value

OR

95%CL

Lower

Upper

1

Vitamin K*sex

0.002

0.004

1.002

1.001

1.004

2

Vitamin K *sex

0.003

0.003

1.003

1.001

1.004

3

Vitamin K *sex

0.002

0.003

1.002

1.001

1.004

4

Vitamin K *sex

0.003

0.003

1.003

1.001

1.004

Significant estimates are in bold. Abbreviations: BMI, body mass index; RS, resilience status; OR, odds ratio, CI, confidence interval. Model 1; adjusted for age and BMI, Model 2; adjusted for age, BMI, current smoker, and current drinker. Model 3; adjusted for age, BMI, education, and occupation, Model 4; adjusted for age, BMI, current smoker, current drinker, education, and occupation.

Reviewer 2 Report

This is an interesting study, given the limited number of studies concerning the relationship between vitamin intake and resilience, understood as the psychological response to environmental stress.

Author Response

Reviewer 2

Comments and Suggestions for Authors

This is an interesting study, given the limited number of studies concerning the relationship between vitamin intake and resilience, understood as the psychological response to environmental stress.

Response

We appreciate your efforts and understanding.
